# ABS: ENFORCING CONSTRAINT SATISFACTION ON GENERATED SEQUENCES VIA AUTOMATA-GUIDED BEAM SEARCH

## ABSTRACT

Sequence generation and prediction form a cornerstone of modern machine learning, with applications spanning natural language processing, program synthesis, and time-series forecasting. These tasks are typically modeled in an autoregressive fashion, where each token is generated conditional on the preceding ones, and beam search is commonly used to balance exploration and fluency during decoding. While deep learning models and Large Language Models (LLMs) excel at capturing statistical patterns in this setting, they remain ill-equipped to guarantee compliance with formal constraints. In this paper, we introduce ABS: a general and model-agnostic inference-time algorithm that guarantees compliance with any constraint that can be compiled into a Deterministic Finite Automaton (DFA), without requiring retraining. ABS leverages the DFA to guide a constrained variant of beam search: at each decoding step, transitions leading to violations are masked, while remaining paths are dynamically re-ranked according to both the model's probabilities and the automaton's acceptance structure. We formally prove that the resulting sequences are guaranteed to satisfy the given constraints, and we empirically demonstrate that ABS also improves output quality. We validate our approach on three distinct tasks: constrained image-stream classification, controlled text generation, and text infilling. In all settings, ABS achieves perfect constraint satisfaction, while outperforming or matching state-of-the-art baselines on standard quality metrics and efficiency.

## 1 INTRODUCTION

Sequence generation is a fundamental paradigm in machine learning, underpinning applications such as natural language processing , program synthesis , and image-stream prediction. These problems are typically approached through autoregressive modeling, where outputs are constructed sequentially, and beam search is a standard decoding strategy used to trade off between exploration and fluency. Despite their remarkable ability to capture statistical patterns in this setting, autoregressive models like Large Language Models (LLMs) offer no guarantees that the sequences they produce will satisfy formal constraints, such as temporal specifications or structural rules.

Existing approaches to constrained sequence generation can be grouped into four main categories: (i) constrain the beam search to ensure the presence or absence of specific outputs (Lu et al., 2021; 2022), (ii) use auxiliary models to steer the sequence generation (Krause et al., 2021; Zhang et al., 2024), (iii) treat constraints as conditioning and sample from the posterior (Miao et al., 2019; Loula et al., 2025), and (iv) employ automata-based guidance to enforce complex constraints (Willard and Louf, 2023; Lundberg et al., 2024; Manginas et al., 2025; Umili et al., 2023). However, none of the existing approaches simultaneously (i) guarantees constraint satisfaction, (ii) avoids additional finetuning or auxiliary models, (iii) achieves low latency, and (iv) preserves the quality of the generated text.

To address these limitations, we propose ABS, a decoding framework that enforces user-specified constraints during autoregressive generation. A key design choice is that our method targets hard, non-negotiable constraints.This makes ABS particularly suited for high-stakes and safety-critical applications. In contrast, soft or probabilistic patterns (e.g., "usually after event A, event B happens") are naturally handled by the probabilistic nature of autoregressive models. Thus, our method is

complementary: the model accounts for soft regularities through its learned distribution, while `ABS` provides an additional mechanism that ensures strict adherence to symbolic rules.

Our method supports any constraint that can be compiled into a Deterministic Finite Automaton (DFA), and therefore applies to the full class of regular languages. This generality subsumes several specification formalisms, including Linear Temporal Logic over finite traces (LTLf) and regular expressions. Consequently, `ABS` is not tied to a single logic, but rather provides a unifying mechanism to enforce diverse temporal and structural rules across modalities.

Starting from a DFA expressing the hard constraints, we integrate its transition dynamics into beam search by re-ranking tokens according to both model logits and DFA state information. This integration guides the generation away from non-accepting sink states (constraint violated) and toward accepting states (constraint satisfied), thereby guaranteeing that all generated sequences satisfy the specified constraints. A natural concern is that such guidance could bias the generation, e.g., by producing overly short outputs or interfering with reasoning. To mitigate this, `ABS` introduces a Ramping Push-Up mechanism: the influence of the automaton is adaptive, starting very gentle when many decoding steps remain and gradually intensifying only if the model risks running out of steps to reach acceptance. This design preserves the model's natural generation abilities while ensuring that constraints are ultimately satisfied.

We demonstrate the broad applicability of `ABS` by evaluating it across three distinct tasks: image sequence classification, constrained text generation, and text infilling. In all settings, `ABS` achieves perfect constraint satisfaction while matching or surpassing state-of-the-art baselines on standard quality metrics. `ABS` achieves these results with substantially lower computational overhead, yielding faster runtimes at fixed beam sizes. Moreover, our text generation experiments show that our adaptive Ramping Push-Up mechanism achieves a careful balance between control and naturalness, making `ABS` offer both formal guarantees and practical flexibility. Our implementation and benchmarks are provided in the supplementary materials and will be released publicly on GitHub.

## 2 NOTATION AND PROBLEM STATEMENT

**Notation.** Let $\mathcal{X}$ be the finite set of possible values that the generated outputs can take at every time step (i.e., our vocabulary). Let $\mathcal{X}^+$ denote the set of all non-empty, finite sequences over $\mathcal{X}$. Let $f_\theta$ be a neural network with learnable weights $\theta$, generating finite sequences of outputs, each denoted by $x_{1:T}$ ($T$ can vary from one sequence to another). Given a sequence $x_{1:T}$, we denote the prefix up to the $t$-th output by $x_{<t}$ and the $t$-th element of the sequence by $x_t$. Finally, let $p_\theta(x_{1:T}) = \prod_{i=1}^{T} p_\theta(x_t \mid x_{<t})$ be the probability of generating the sequence $x_{1:T}$, where $p_\theta(x_t \mid x_{<t})$ is defined by applying a softmax to the logits outputted by $f_\theta(x_{<t})$.

**Definition 1.** *A Deterministic Finite Automaton (DFA) is a 5-tuple $\mathcal{A} = (\mathcal{Q}, \mathcal{C}, \delta, q_0, \mathcal{F})$ where: (i) $\mathcal{Q}$ is a finite set of states, (ii) $\mathcal{C}$ is a finite set of symbols (the alphabet), (iii) $\delta : \mathcal{Q} \times \mathcal{C} \to \mathcal{Q}$ is the transition function, (iv) $q_0 \in \mathcal{Q}$ is the initial state, (v) $\mathcal{F} \subseteq \mathcal{Q}$ is the set of accepting states.*

An example of a DFA is given in Figure 1 (left). In simple cases, the DFA alphabet $\mathcal{C}$ coincides with the model vocabulary $\mathcal{X}$. However, our approach supports more complex mappings. For example, LLMs typically output tokens while constraints are often written over words. To account for such cases, , we assume there exists an injective function $\nu : \mathcal{X}^+ \to \mathcal{C}^+$ such that $\nu(x_{1:T}) = c_{1:m}$. This function maps a raw output sequence from the network (e.g., a sequence of tokens) to a sequence of higher-level concepts (e.g., words) that form the input symbols for the DFA.

A DFA $\mathcal{A}$ accepts a sequence $c_{1:m} = \nu(x_{1:T}) \in \mathcal{C}^+$, noted $\mathcal{A} \vdash c_{1:T}$, if and only if there exists a sequence of states $r_0, r_1, \ldots, r_m \in \mathcal{Q}^+$ such that: $r_0 = q_0$, $r_i = \delta(r_{i-1}, c_i)$ for $i = 1, \ldots, m$ and $r_m \in \mathcal{F}$. In addition, a DFA state is a *deadlock state* or *sinking state* if it is a non-accepting state from which no accepting state is reachable via any sequence of transitions.

**Problem Statement.** Given $f_\theta$ and a DFA $\mathcal{A}$, we target the problem of ensuring $\mathcal{A} \vdash \nu(x_{1:T})$ for every sequence $x_{1:T}$ generated by $f_\theta$, while maximizing the sequence log-likelihood under $p_\theta$, i.e.,

$$x_{1:T}^\star = \underset{x_{1:T} \in \mathcal{X}^T}{\mathrm{argmax}} \sum_{t=1}^{T} \log p_\theta(x_t \mid x_{<t}) \quad \text{such that } \mathcal{A} \vdash \nu(x_{1:T}). \tag{1}$$

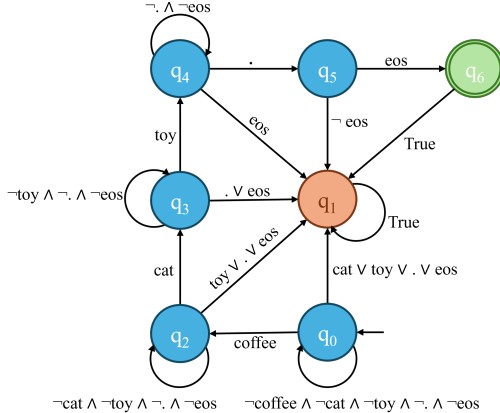

| State | coffee | cat | toy | . | eos |
|-------|--------|-----|-----|---|-----|
| $q_0$ | $q_3$ | $q_2$ | $q_2$ | $q_2$ | $q_2$ |
| $q_1$ | $q_2$ | $q_2$ | $q_2$ | $q_2$ | $q_3$ |
| $q_2$ | $q_3$ | $q_4$ | $q_2$ | $q_2$ | $q_2$ |
| $q_3$ | $q_4$ | $q_4$ | $q_5$ | $q_2$ | $q_2$ |
| $q_4$ | $q_5$ | $q_5$ | $q_5$ | $q_6$ | $q_2$ |
| $q_5$ | $q_2$ | $q_2$ | $q_2$ | $q_2$ | $q_7$ |
| $q_6$ | $q_2$ | $q_2$ | $q_2$ | $q_2$ | $q_2$ |

**Distances to closest accepting state:**

$$d(q_0) = 5 \quad d(q_1) = +\infty \quad d(q_2) = 4$$
$$d(q_3) = 3 \quad d(q_4) = 2 \quad \quad d(q_5) = 1$$
$$d(q_6) = 0$$

Figure 1: DFA (left) and corresponding state transition table with the distances to the closest accepting states (right) for the constraint: "*Generate a sentence that contains coffee, cat, and toy in that order and ends with a dot*" (note how this constraint can be easily expressed as a regex or LTL$_f$ formula). The deadlock state is $q_1$, the accepting state is $q_6$ while the initial state is $q_0$.

This corresponds to the *Maximum A Posteriori* (MAP) inference problem under constraints specified by the DFA $\mathcal{A}$.

## 3 THE ABS ALGORITHM

The ABS algorithm consists of two main steps: (i) the *Automata Preprocessing Step*, where we process the given DFA into an efficient computational representation and (ii) the dynamic *Automata-guided Beam Search*, where the beam search is guided by both the model's predictions and the DFA.

### 3.1 AUTOMATA PREPROCESSING STEP

In our setting, the maximum length $T$ of the output sequence is not known at the time of DFA preprocessing and is only provided at runtime. Therefore, our automaton processing step must be prepared to handle traces of any finite length.

First, we process the given DFA $\mathcal{A} = (\mathcal{Q}, \mathcal{C}, \delta, q_0, \mathcal{F})$ into a more computationally efficient form. Hence, we represent the transition function $\delta$ as a matrix $\mathbf{M} \in \mathbb{Z}^{|\mathcal{Q}| \times |\mathcal{C}|}$, where each entry $m_{q,c}$ stores the next state $q' = \delta(q, c)$ for state $q \in \mathcal{Q}$ and symbol $c \in \mathcal{C}$. For notational convenience, we use $q$ and $c$ to denote both states/concepts and their corresponding indices in $\mathbf{M}$. This matrix representation enables efficient state transition lookups during the generation process.

However, the network $f_\theta$ might not necessarily output symbols in the alphabet $\mathcal{C}$. Hence, we also create the cost function $w : \mathcal{C} \to \mathbb{N}$ representing the number of outputs $f_\theta$ has to generate to create concept $c$. This will also represent the cost to move from $q$ to $q' = \delta(q, c)$. How the cost function is defined varies from one use case to another. For example, in constrained text generation, if the DFA is defined over tokens (i.e., each symbol in $\mathcal{C}$ corresponds to a single token), then each transition naturally has a cost of 1. On the other hand, if the DFA is defined over higher-level concepts such as words or phrases, each of which might require multiple tokens to generate, then the cost of a transition labeled with a concept $c \in \mathcal{C}$ is set to the minimum number of tokens needed to output $c$.

Given a cost function $w : \mathcal{C} \to \mathbb{N}$ and a DFA $\mathcal{A} = (\mathcal{Q}, \mathcal{C}, \delta, q_0, \mathcal{F})$, we define the corresponding *weighted DFA* $\mathcal{A}^w$ by assigning each transition $\delta(q, c)$ a cost $\tilde{w}(q, c) = +\infty$ if $q$ is a sinking state and $\tilde{w}(q, c) = w(c)$ otherwise. A sequence $c_{1:m} \in \mathcal{C}^*$ is accepted by $\mathcal{A}^w$ iff it is accepted by $\mathcal{A}$. For any sequence $c_{1:m}$, its cost is defined as

$$W(c_{1:m}) = \sum_{i=1}^{m} \tilde{w}(q_{i-1}, c_i), \tag{2}$$

---

**Algorithm 1** ABS algorithm with Ramping Push-Up

---

1: **Input:** Prompt $x_0$; token set $\mathcal{X}$; LLM log-prob function $\log p_\theta$; DFA $(\mathcal{Q}, \mathcal{C}, \delta, q_0, \mathcal{F})$; distance-to-accepting-state function $d$; num. beams $k$; max length $T$; ramping params $(\alpha_{\min}, \gamma)$
2: **Output:** DFA compliant sequence $\hat{x}_{1:T}$
3:
4: Initialize $k$ beams $\mathcal{B} \leftarrow \{(x_0, 0, q_0, d(q_0), \epsilon), \ldots, (x_0, 0, q_0, d(q_0), \epsilon)\}$    ▷ $\epsilon$ indicates empty string
5: **for** $t = 1$ to $T$ **do**
6:     Extend logits for all beams: $\mathbf{Z} \leftarrow \left[ \log p_\theta(x \mid x_{<t}^i) \right]_{i \in [k], x \in \mathcal{X}} \in \mathbb{R}^{k \times |\mathcal{X}|}$
7:     $\mathcal{S} \leftarrow \emptyset$                                                         ▷ Candidates pool
8:     **for** $i = 1$ to $k$ **do**
9:         $\alpha_t^i \leftarrow \text{RAMPPUSHUP}(\alpha_{\min}, d_t^i, t, T, \gamma)$                    ▷ See Equation 3
10:         **for all** $x \in \mathcal{X}$ **do**
11:             $q' \leftarrow \text{NEXTSTATE}(q_t^i, l_t^i, x)$                              ▷ See Equation 4
12:             **if** $d(q') > T - t$ **then**
13:                 **continue**                       ▷ Skip: we cannot reach acceptance in $(T - t)$ steps
14:             **end if**
15:             $\tilde{z} \leftarrow \begin{cases} \alpha_t^i \cdot \max \mathbf{Z}[i, :] + (1 - \alpha_t^i) \cdot \mathbf{Z}[i, x], & \text{if } d(q') < d_t^i \\ \mathbf{Z}[i, x], & \text{otherwise} \end{cases}$
16:             $s' \leftarrow s^i + \tilde{z}$
17:             $\mathcal{S} \leftarrow \mathcal{S} \cup \{(x_{<t}^i x, s', q', d(q'), l_t^i x)\}$ ▷ $x_{<t}^i x, l^i x$ are new sequences obtained via concatenation
18:         **end for**
19:     **end for**
20:     $\mathcal{B} \leftarrow \text{TOPK}(\mathcal{S}, k; \text{ key} = s_t^i)$                        ▷ Pick the $k$ best successors by score
21: **end for**
22: **return** Token sequence $\hat{x}_{1:T} = x_{1:T}^{i^\star}$, where $i^\star = \arg\max s_T^i$

---

i.e., the sum of the transition costs along the unique accepting run. We apply Dijkstra's algorithm (Dijkstra, 1959) to the weighted DFA in order to compute, for each state $q$, the minimal cost of reaching an accepting state $q \in \mathcal{F}$. We define a distance function $d : \mathcal{Q} \to \mathbb{N} \cup \{+\infty\}$, where $d(q)$ gives the minimal cost of reaching an accepting state from $q$.

## 3.2 AUTOMATA-GUIDED BEAM SEARCH

Automata-guided Beam Search has two goals: (i) prevent beams from entering deadlock states (from which no accepting state is reachable), and (ii) bias exploration toward accepting states. The full procedure is detailed in Algorithm 1, with a graphical example in Figure 2. Throughout this Section and in the Algorithm, given two sequences $x_1, x_2$, we will indicate with $x_1 x_2$ their concatenation.

In the Algorithm, for every $t$ we maintain a set $\mathcal{B}$ of $k$ tuples, each representing the state of one beam. The $i$-th tuple has form $(x_{<t}^i, s_t^i, q_t^i, d_t^i, l_t^i)$, where: (i) $x_{<t}^i$ is the partial sequence generated so far, (ii) $s_t^i$ is the score of the beam which we will compute taking into account both $\mathbf{Z}$ and the DFA, (iii) $q_t^i$ is the current DFA state, (iv) $d_t^i = d(q_t^i)$ is the distance from $q_t^i$ to the closest accepting state, and (v) $l_t^i$ is a sequence of outputs representing the last concept being "built up" to move to the next state (possibly still not completed, e.g., if the LLM is trying to generate "politician", $l_t^i$ might be "polit"). At $t = 0$, all the tuples in $\mathcal{B}$ are initialized in the same way as $(x_0, 0, q_0, d(q_0), \epsilon)$, $x_0$ being the initial prompt and $\epsilon$ being the empty sequence.

**Ramping Push-Up.** We want the model mostly to follow its own distribution, but to gradually steer it to avoid dead ends and force it to satisfy the constraints when necessary. Hence, we introduce a dynamic mechanism that biases decoding toward DFA transitions leading closer to an accepting state. At each step $t$ and for each beam $i$, we define a coefficient $\alpha_t^i \in [\alpha_{\min}, 1]$ that scales how strongly we push the logits of promising outputs toward the maximum logit. The coefficient increases as the remaining steps $T - t$ approach the current DFA distance to acceptance $d_t^i$:

$$\alpha_t^i = \text{RAMPPUSHUP}(\alpha_{\min}, d_t^i, t, T, \gamma) \overset{\text{def}}{=} \alpha_{\min} + (1 - \alpha_{\min}) \cdot \min\left(1, \left(\frac{d_t^i}{T - t}\right)^\gamma\right), \quad (3)$$

Figure 2: **(Cont.'ed from Figure 1)**. Real generation using ABS, with 2 beams, $\alpha_{\min}$=0.5, $\gamma$=1, and T=10. The selected sequence is shaded in green. Above each token are the relative logits, before and after modification, while the arcs represent the state transitions inside the automaton. The same model without ABS generates "*The cat was playing with a toy while the owner*", breaking the order constraints on word generation (cat before coffee). After generating "The", the model prefers "cat" at $t = 2$, violating the requirements.

where $\gamma > 0$ controls the sharpness of the ramp. This schedule encourages natural generation when slack is high ($T - t \gg d_t^i$), as the bias is mild ($\alpha_t^i \approx \alpha_{\min}$), and only intensifies when the model risks running out of steps ($T - t \approx d_t^i$). Additionally, the sharpness parameter $\gamma$ controls how quickly the pressure ramps up as the sequence nears its limit, with smaller values yielding smoother guidance and larger values enforcing a sharper transition. In practice, both $\alpha_{\min}$ and $\gamma$ are selected empirically on the validation set.

**Next State.** The DFA transitions are defined over symbols in $\mathcal{C}$, while the network $f_\theta$ (and hence the beam search) operate over the set of output symbols $\mathcal{X}$. To overcome this limitation, given the current state $q$, the sequence of outputs representing the last concept being built up $l$, and the current candidate output $x$:

$$\text{NEXTSTATE}(q, l, x) = \begin{cases} \delta(q, c') & \text{if } c' = lx \in \mathcal{C}, \\ q & \text{if } \exists c \in \mathcal{C} \text{ with prefix } lx, \\ \delta(q, \text{NOMATCH}) & \text{otherwise.} \end{cases} \quad (4)$$

The symbol NOMATCH is a special input that preserves DFA consistency when $x$ does not extend any concept, allowing for unconstrained outputs (e.g., filler words) while preserving DFA consistency.

**Distance from Accepting State.** Before adding a candidate to the set of effective candidates, we verify that its distance to the nearest accepting state is at most $T - t$ (line 13 of Algorithm 1). If this condition is not met, the candidate is skipped. This check prevents the model from entering a state from which it cannot reach an accepting state within the remaining steps. Combined with the RPU mechanism, this process guarantees that all constraints are satisfied, as proven in Theorem 1.

## 4 THEORETICAL ANALYSIS

In this section we provide formal statements concerning the computational complexity of our inference tasks, soundness of our Automata-guided inference method, and comparison with existing works. Proofs of the stated theorems are reported in the Appendix.

### 4.1 SOUNDNESS AND COMPLEXITY OF AUTOMATA-GUIDED BEAM SEARCH

We recall that our problem (Eq. 1) is a constrained Maximum A Posteriori (MAP) inference task over the outputs of an autoregressive model. MAP inference with logical constraints is NP-hard in general, even for Bayesian networks of treewidth one (Roth, 1996). We therefore design a structured approximate method that is tractable in practice yet guarantees soundness.

**Theorem 1** (Soundness of Automata-guided Beam Search with Ramping $\alpha$). *Let $\phi$ be a constraint over a sequence of concepts compiled into a DFA $\mathcal{A}$, and let $q_0$ be the initial state of $\mathcal{A}$. If (i) at step 0, $q_0$ is within distance $d_0 \leq T$ of an accepting state, where $T$ is the maximum sequence length; (ii) at each step $t$, the scheduler ramps $\alpha_t^i$ and transitions to states with $d_{q_{t+1}} > T - t$ are pruned,*

*then the algorithm returns a sequence $\hat{x}_{1:T}$ such that $\hat{x}_{1:T} \models \phi$, whenever such a sequence exists.*

This guarantees that Automata-guided Beam Search never discards all feasible paths: if a satisfying sequence exists, it will be produced.

**Complexity** For a fixed-size DFA, Automata-guided Beam Search runs in polynomial time with respect to sequence length $T$, beam width $k$, and output space size $|\mathcal{X}|$. Each decoding step considers at most $k \cdot |\mathcal{X}|$ candidates, applies DFA filtering and scoring, and retains the top $k$. The overall time complexity is therefore

$$O(T \cdot k \cdot |\mathcal{X}|).$$

### 4.2 Theoretical Comparison with the State of the Art

One method to address this problem, called Ctrl-G, and proposed by Zhang et al. (2024), is to distill the model $f_\theta$ into Markov models to hopefully obtain a tractable representation. Unfortunately, the next result shows that even in the simplest case, the constrained MAP inference task is NP-hard.

**Theorem 2** (Complexity of Markov Chains MAP Inference with Unary Constraint). *MAP inference on Markov Chains with a unary equality constraint is NP-hard.*

Thus, while HMMs admit polynomial-time MAP inference via Viterbi under dense emissions, LLM outputs are typically sparse, so HMM approximations (Zhang et al., 2024) might not guarantee tractable constrained decoding. Finally, we note that other Automata-based steering methods, including industry standards like Guidance (Lundberg et al., 2024) and Outlines (Willard and Louf, 2023), are shown to be unsound through experimental results in the following section.

## 5 Experimental Evaluation

We report on experiments assessing the benefits of our approach applied to three tasks: image sequence classification and text generation, involving temporal constraints (expressed as LTL$_f$ formulas), and text infilling, involving structural constraints (expressed as regular expressions). To produce the automata representing the LTL$_f$ formulae, we exploit MONA [1] and LTLf2DFA [2] (Fuggitti, 2019). For regular expressions, we used FAdo (Reis and Moreira, 2002). All experiments ran on a machine with an Intel® Xeon® 20 cores and NVidia L40S GPU with 48 GB of VRAM.

### 5.1 Constrained Image Sequence Classification

We first evaluate our method on a controlled sequential image classification task. Starting from Fashion-MNIST[3], we construct *Ordered Fashion-MNIST*, in which images are arranged into sequences subject to LTL$_f$ constraints we manually annotated (Appendix C). The constraints specify which clothing items cannot co-occur and enforce ordering constraints. For example, the sequence [trousers, t-shirt/top, sneakers, sandals] is not allowed as sneakers and sandals cannot be worn at the same time, but neither is the sequence [sandals, trousers,

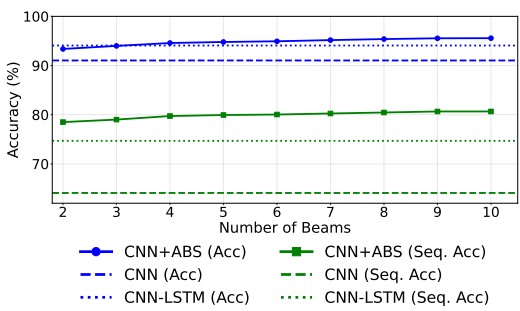

Figure 3: Performance on Ordered Fashion MNIST.

[1]https://www.brics.dk/mona/, copyright 1997-2020 Aarhus University.

[2]http://ltlf2dfa.diag.uniroma1.it/, License LGPLv3+

[3]https://github.com/zalandoresearch/fashion-mnist, MIT License

Table 1: Performance on CommonGen. `ABS` is run with: number of beams=64, $\alpha$=0.5 and $\gamma$=1, for both supervised and unsupervised model. Ctrl-G uses the HMM with 32768 hidden states the authors provided in their paper. Best scores are in bold, while second bests are underlined.

| Method | ROUGE-L | BLEU-4 | CIDEr | SPICE | Coverage |
|---|---|---|---|---|---|
| *Unsupervised* | | | | | |
| InsNet (Lu and Peng (2021)) | - | 18.7 | - | - | **100.0** |
| NADO (Meng et al. (2022)) | - | 26.2 | - | - | 96.1 |
| GeLaTo (Zhang et al. (2023)) | 44.3 | 30.3 | 15.6 | 30.2 | **100.0** |
| Ctrl-G (Zhang et al. (2024)) | 45.2 | 32.1 | **16.0** | **30.8** | **100.0** |
| Outlines (Willard and Louf (2023)) | 31.4 | 18.7 | 2.8 | 19.7 | 80.6 |
| Guidance (Lundberg et al. (2024)) | 19.4 | 9.2 | 3.3 | 15.3 | 92.1 |
| **ABS** | **48.7** | **47.9** | 15.7 | 29.3 | **100.0** |
| *Supervised* | | | | | |
| NADO | 44.4 | 30.8 | 16.1 | 32.0 | 88.8 |
| GeLaTo | 46.2 | 34.0 | **17.2** | **32.2** | **100.0** |
| Outlines | 29.9 | 16.3 | 2.7 | 18.5 | 76.5 |
| Guidance | 21.3 | 9.9 | 3.0 | 15.3 | 89.0 |
| **ABS** | **49.7** | **49.5** | 16.2 | 29.7 | **100.0** |

Table 2: Runtimes comparison across different beam sizes ($\pm$ standard error), with max length $T = 32$. `ABS` is used with $\alpha$=0.5 and $\gamma$=1. Ctrl-G uses the HMM with 32768 hidden states. Both use fine-tuned GPT2.

| Method | 4 | 8 | 16 | 32 | 64 | 128 |
|---|---|---|---|---|---|---|
| Ctrl-G | $3.49 \pm 0.011$ | $3.70 \pm 0.014$ | $4.25 \pm 0.024$ | $5.43 \pm 0.053$ | $8.05 \pm 0.102$ | $12.01 \pm 0.161$ |
| **ABS** | $\mathbf{0.74} \pm 0.007$ | $\mathbf{0.81} \pm 0.006$ | $\mathbf{1.33} \pm 0.010$ | $\mathbf{2.22} \pm 0.019$ | $\mathbf{4.42} \pm 0.041$ | $\mathbf{9.45} \pm 0.084$ |

t-shirt/top] as sandals cannot be worn before trousers.

As a baseline, a frozen CNN trained on Fashion MNIST achieves 91.03% accuracy and 64.09% sequence accuracy (i.e., the ratio of sequences that are correctly classified) on Ordered Fashion-MNIST. As an additional baseline, we combined the frozen CNN with a trainable LSTM, which allows us to model sequential dependencies. In this setting, only the LSTM parameters are trained, while the CNN remains fixed. This CNN-LSTM sequence model achieves an accuracy of 94.07% and a sequence accuracy of 74.69%. In contrast, combining the frozen CNN with `ABS` (10 beams) yields 95.56% accuracy (+4.53 with respect to the CNN and +1.49 to the CNN-LSTM) and 80.66% sequence accuracy (+16.57 with respect to the CNN and +5.97 to the CNN-LSTM), surpassing both baselines without additional training. Figure 3 shows how performance improves with beam size. This experiment shows that our method replaces the need for such an additional recurrent module, avoiding extra training, while achieving better performance, especially in the sequence accuracy. Implementation details are provided in Appendix C.

## 5.2 CONSTRAINED TEXT GENERATION

We present two experiments on constrained text generation with LLMs. For both datasets, we computed the standard quality metrics in the literature: BLEU (Papineni et al., 2002), ROUGE (Lin, 2004), CIDEr (Vedantam et al., 2015) and SPICE (Anderson et al., 2016).

**CommonGen.** For the first experiment we use the CommonGen dataset (Lin et al., 2020), a standard evaluation task for constrained text generation released under MIT License. In this task, a set of common concepts is provided (e.g., dog, frisbee, catch, throw) and the goal for the LLM is to generate a coherent sentence describing a scenario using all the given concepts (e.g., "A man throws a frisbee, and his dog catches it"). We compare our results with those obtained in relevant previous work, such as works of Zhang et al. (2024; 2023); Meng et al. (2022). In particular, we use the scores reported

Table 3: Performances on Ordered CommonGen. `ABS` use LLAMA 3.1 8B with number of beams=64, $\alpha$=0.25, $\gamma$=1. The best scores are in bold, while the second best are underlined.

| Method/Model | ROUGE-L | BLEU-4 | CIDEr | SPICE | Coverage |
|---|---|---|---|---|---|
| GPT 3.5 | 42.2 | 24.9 | 13.5 | 40.8 | 64.5 |
| GPT 4 | 42.2 | 23.7 | 12.4 | 40.7 | 83.3 |
| GPT 4o | 42.7 | 25.2 | 13.4 | 42.2 | 73.9 |
| o1 | **43.1** | 24.3 | 12.9 | 41.3 | 99.8 |
| LLAMA 3.1 8B | 32.2 | 22.6 | 12.3 | 38.0 | 44.4 |
| LLAMA 3.1 8B + Outlines | 33.5 | 21.8 | 10.4 | 40.5 | 49.9 |
| LLAMA 3.1 8B + Guidance | 38.0 | 22.8 | 10.4 | **43.0** | 54.0 |
| LLAMA 3.1 8B + **ABS** | 40.0 | **25.6** | **13.6** | 41.7 | **100.0** |

in the first two cited papers. Moreover, to ensure a fair comparison, we apply `ABS`, Guidance and Outlines to the same models they employed: GPT2-large (Radford et al., 2019) (MIT License) as the unsupervised model, and GPT2-large fine-tuned by Zhang et al. (2023) on the training set of CommonGen as the supervised model. For this task we selected the best parameters on the training set and then applied these parameters to the validation set.

The results in Table 5.1 show that, for both models, `ABS` exceeds the current state of the art in two quality metrics (ROUGE-L and BLEU-4), is competitive in the remaining two (CIDEr and SPICE) and achieves 100% constraint satisfaction ("Coverage"). Moreover, these experiments show that approaches such as Outlines and Guidance, which also rely on DFAs to guide generation, do not achieve 100% coverage. This is because these methods prevent entering a deadlock state (i.e. constraint is violated) but they lack a mechanism that guarantees to land in an accepting state (i.e. constraint is satified) before reaching the model's maximum token limit. Importantly, as shown in Table 2, `ABS` is systematically faster than Ctrl-G – the best competing method – when using the same number of beams and maximum token generation length.

**Ordered CommonGen.** In our second experiment, we introduce Ordered CommonGen, a variant of CommonGen with a temporal constraint that the generated sentence must include all provided concepts in a specified order. For example, given the ordered set *dog, frisbee, catch, throw*, a valid output would be: "The dog eagerly chased the frisbee trying to catch it after its owner threw it.". This ordering constraint increases task difficulty and makes the setting well-suited for evaluating temporal constraint satisfaction. The prompt adapted from CommonGen Lite, is provided in Appendix D.1.

We compare `ABS` applied to LLAMA 3.1 8B with Guidance and Outlines applied to the same model, and with large OpenAI models: GPT-3.5, GPT-4, GPT-4o, and the o1 reasoning model Brown et al. (2020); OpenAI (2024a;b;c). Table 3 shows that, despite using a much smaller and locally deployed model, `ABS` outperforms the large models on two quality metrics (BLEU-4 and CIDEr) and remains competitive on the other two metrics. It is also the only approach that achieves 100% constraint satisfaction, consistent with its sound design, whereas GPT-3.5, GPT-4, and GPT-4o have a significant number of violations and o1 achieves 99.8% coverage. Compared to the base LLAMA 3.1 8B model, `ABS` achieves a substantial improvement in qualitative metrics (R: +7.8, B: +3.0, C: +1.3, S: +3.7, and Cov.: +55.6). While Guidance and Outlines also improve performance on most metrics, they do not reach the improvements made by `ABS` (except Guidance on SPICE metric only) and they yield significantly lower coverage than the various GPT models and `ABS`.

## 5.3 TEXT INFILLING

We also evaluate `ABS` on a text infilling benchmark introduced by Donahue et al. (2020), which is based on the ROC Stories corpus (Mostafazadeh et al. (2016)). Each test example consists of a short story with masked segments, and the task is to fill in these masks. For instance: *"My day on <|infill_word|> this week went as expected. My family and I attended Church. <|infill_sentence|>"*. We straightforwardly express these text infilling tasks with regular expressions.

We use the GPT-2 small checkpoint released by Donahue et al. (2020) as part of their ILM method, as the base model for `ABS`, Guidance and Outlines. Following their experimental protocol, we generate three test sets from the original one, with masking ratios of 10%, 20%, and 30%. To evaluate the

Table 4: Performances on Text Infilling. `ABS` is run with: with number of beams=16, $\alpha$=0.5, $\gamma$=1. Best scores are in bold, while second bests are underlined.

| Method | 10% | | | 20% | | | 30% | | |
|---|---|---|---|---|---|---|---|---|---|
| | Rouge-L | Bleu-4 | Cov. | Rouge-L | Bleu-4 | Cov. | Rouge-L | Bleu-4 | Cov. |
| ILM | 74.5 | 71.7 | 86.5 | 63.2 | 59.8 | 49.7 | 51.1 | 46.5 | 23.0 |
| Guidance | 80.5 | 70.8 | 68.0 | 67.9 | 56.1 | 56.0 | 55.6 | 42.4 | 36.0 |
| Outlines | 71.2 | 56.4 | 94.0 | 55.7 | 38.8 | 80.0 | 43.6 | 28.6 | 80.0 |
| **ABS** | **85.4** | **79.1** | **100.0** | **73.3** | **64.1** | **100.0** | **61.6** | **50.3** | **100.0** |

completed stories, we report BLEU-4 and ROUGE-L and Coverage. As shown in Table 4, `ABS` outperforms ILM, Guidance, and Outline. `ABS` is again the only method to achieve perfect coverage.

## 6 RELATED WORK

**Neurosymbolic AI.** Neurosymbolic AI is a growing filed of interest in machine learning due to the ability of its methods to reconcile neural perception with symbolic reasonings (Raedt et al., 2020; d'Avila Garcez and Lamb, 2023). While the field is quite vast, the neurosymbolic solutions that are the closest to our method are those that try to incorporate symbolic background knowledge in the form of constraints into neural models (Dash et al., 2022; Giunchiglia et al., 2022). These methods can be divided in two groups: those that give a probabilistic semantics to the constraints (Manhaeve et al., 2018; Xu et al., 2018; van Krieken et al., 2023) and those that give a fuzzy semantics (Donadello et al., 2017; Giunchiglia et al., 2024; Diligenti et al., 2012). Another way to categorize the methods in the field is by whether they incorporate the constraints in the loss function (Donadello et al., 2017; Xu et al., 2018; Fischer et al., 2019; van Krieken et al., 2022) or they change the final output space of the model (Manhaeve et al., 2018; Giunchiglia and Lukasiewicz, 2021; Ahmed et al., 2022; Pryor et al., 2023; Hoernle et al., 2022; Misino et al., 2022). The first can be used to nudge via the loss the network to satisfy the constraints, while the second can guarantee the constraints satisfaction.

**Constrained Text Generation.** The work done in constrained text generation has been developing on multiple orthogonal axes. *1. Search based decoding.* These approaches act at inference time and constrain the beam search with logical constraints. For example, in (Lu et al., 2021; 2022), the authors decide which beams to expand taking into account not only the LLM's prediction but also the degree of satisfaction of the constraints reported for every sequence. *2. Auxiliary Classifier Guidance.* The works that follow this line aim to guide the base model with an auxiliary one. GeDi (Krause et al., 2021), FUDGE (Yang and Klein, 2021) and NADO (Meng et al., 2022) steer generation with class-conditional or token-level predictors. However, they provide no hard guarantees and might require task-specific supervision. On the contrary, GeLaTo (Zhang et al., 2023) provides the guarantee as it distills the base LLM into a hidden-Markov model, which computes the probability that the remaining suffix can still contain a set of keywords and then multiplies this value into the LLM's logits. Ctrl-G (Zhang et al., 2024) is the recent extension of GeLato, which allows for any constraint that can be compiled to a DFA (instead of propositional logic). *3. Probabilistic Sampling* treats constraints as conditioning and samples from the posterior. This line was pioneered by (Miao et al., 2019) with Metropolis–Hastings token edits that provide unbiased samples under lexical constraints. This idea has then been further explored in the works of Ahmed et al. (2025); Loula et al. (2025).

## 7 CONCLUSIONS

We introduced `ABS`, a DFA-guided method to constrain the output of a Neural Network. Because we enforce specifications via a compiled DFA, any *regular* constraint (e.g., regex or LTL$_f$) is supported, enabling both temporal and structural requirements, and thus broadening the range of case studies w.r.t. existing methods. Our algorithm is proved to be sound, while not optimal, and can easily be adapted to different applications. Comparison with the state of the art shows significant improvements in both efficiency and quality of the generated output. The introduced method can be used for multiple purposes, with a positive societal impact, like the detoxification of LLMs' output.

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

## A  PROOFS

### A.1  SOUNDNESS OF THE ABS ALGORITHM

We prove here that our method is sound (Theorem 1).

*Proof of Theorem 1. (Soundness of the ABS algorithm).* We prove that the algorithm is sound by induction on the length $t$ of the sequence. More precisely, we propose the following inductive hypothesis

$$\mathcal{H}_t : \text{for all beams } (x_{<t}, s, q) \in \mathcal{B}_t, d_{q_t} \leq T - t.$$

The base case is $t = 0$: the initial beam only contains the prompt $x_0$, the initial state of the DFA $q_0$, and the distance $d_0$ from an accepting state. By the hypothesis of the theorem, we have $d_0 \leq T$ and thus $\mathcal{H}_0$ is verified.

Let us now assume that $\mathcal{H}_t$ is true, for $t \in [1, T - 1]$. At line 12 of the algorithm:

- if a transition leads to a state $q'$ such that $d_{q'} > T - t$ then it is impossible to complete an accepting sequence within the remaining steps. But this choice is excluded by setting $Z[i, x] \leftarrow -\infty$.

This ensures that no sequence in $\mathcal{B}$ is extended to a non-accepting sequence.

Indeed, at line 22:

- we choose the top $k$ beam extensions among the valid ones with respect to their score (denoted by $Z$ in the pseudo-code), i.e. among those having distance $d_{q'_{t+1}} \leq T - (t + 1)$, since invalid outputs have $-\infty$ as a score. In the case where we have less than $k$ valid beam extensions, then we pad the remaining beams with copies of the valid beams. We know that at least one valid extension exists for each beam, since we assume that $\mathcal{H}_t$ is true.

Thus we have shown that $\mathcal{H}_{t+1}$ is true whenever $\mathcal{H}_t$ is true. By induction, $\mathcal{H}_T$ is true.

At time $T$, the algorithm returns the best sequence among those included in $\mathcal{B}_T$. But, by $\mathcal{H}_T$, every sequence in $\mathcal{B}_T$ has distance 0 from an accepting state, and hence terminates in an accepting state. Thus the algorithm is sound. $\square$

### A.2  COMPLEXITY OF MAP INFERENCE ON MARKOV CHAINS

*Proof of Theorem 2.* We show NP-hardness by providing a polynomial-time reduction from the *Constrained Shortest Path (CSP) problem with uniform arc lengths* to the *MAP inference problem on time-homogeneous Markov chains under a unary constraint.*

**Step 1: CSP Formulation.** Consider a directed graph $G = (V, E)$, where each arc $(i, j) \in E$ has an associated cost $c(i, j) \geq 0$ and a length $b(i, j) \in \mathbb{N}$. We focus on the special case where all arcs have the same length, specifically $b(i, j) = 1$ for all $(i, j) \in E$. Given vertices $s, t \in V$ (source and target) and an integer bound $\beta \geq 1$, the CSP problem seeks a path of exact length $\beta$ from $s$ to $t$ that minimizes total cost:

$$\min_{x_{1:\beta} \in V^\beta, \, x_1 = s, \, x_\beta = t, \, (x_k, x_{k+1}) \in E} \sum_{k=1}^{\beta-1} c(x_k, x_{k+1}).$$

This uniform-length CSP variant is known to be NP-hard (Erzin et al., 2022).

**Step 2: MAP Inference Problem Formulation.** In the MAP inference setting, we have a time-homogeneous Markov chain over a state space $S$, an initial distribution $\mu$, and transition matrix $P$. Given a unary constraint specifying the state at time $\beta$, i.e., $X_\beta = t$, the MAP inference problem is:

$$x_{1:\beta}^* = \underset{x_{1:\beta} \in S^\beta, \, x_\beta = t}{\arg\max} \mu(x_1) \prod_{k=1}^{\beta-1} P(x_{k+1} \mid x_k).$$

**Step 3: Reduction from CSP to MAP Inference.** Given a CSP instance $(G, c, s, t, \beta)$, construct a MAP inference instance as follows:

1. **State space.** Define the state space as $S = V \cup D$, where $D = \{d_i \mid i \in V\}$ is a set of newly introduced dummy states, with one dummy state per original vertex.

2. **Initial distribution.** Set $\mu(x) = \delta_{x,s}$, ensuring the chain always starts at the source vertex $s$.

3. **Transition probabilities.** First, define an extended cost function $\tilde{c} : V \times S \to [0, +\infty]$ by:

$$
\tilde{c}(i, j) = \begin{cases} c(i, j), & (i, j) \in E \\ -\log(Z_{\max} - Z_i), & j = d_i \\ +\infty, & \text{otherwise} \end{cases}
$$

where for each vertex $i \in V$, we set:

$$
Z_i = \sum_{j \in V} e^{-c(i,j)}, \quad Z_{\max} = \max_{i \in V} Z_i.
$$

With these definitions, we set the transition probabilities for all $i \in V$ as:

$$
P(j \mid i) = \frac{e^{-\tilde{c}(i,j)}}{Z_{\max}}, \quad j \in S.
$$

This ensures that all transitions from states in $V$ are properly normalized, as:

$$
\sum_{j \in S} P(j \mid i) = \frac{Z_i + (Z_{\max} - Z_i)}{Z_{\max}} = 1, \quad \forall i \in V.
$$

4. **Absorbing states.** Explicitly set the target vertex $t$ and all dummy states $d_i$ as absorbing states:

$$
P(t \mid t) = 1, \quad P(j \mid t) = 0 \text{ for } j \neq t, \quad \text{and} \quad P(d_i \mid d_i) = 1, \quad P(j \mid d_i) = 0 \text{ for } j \neq d_i.
$$

Thus, any path entering a dummy state or $t$ remains there indefinitely.

5. **Unary constraint.** Set the unary constraint as $\phi \equiv X_\beta = t$.

**Step 4: Correctness and Equivalence.** Consider any path $x_{1:\beta}$ that satisfies the constraint $X_\beta = t$. If the path ever enters a dummy state $d_i$, it will remain there indefinitely (absorbing), contradicting the constraint $X_\beta = t$. Thus, any feasible path must stay entirely within $V$.

For feasible paths $x_{1:\beta} \in V^\beta$, we have:

$$
P(x_{1:\beta}) = \prod_{k=1}^{\beta-1} \frac{e^{-\tilde{c}(x_k, x_{k+1})}}{Z_{\max}} = Z_{\max}^{-(\beta-1)} \cdot e^{-\sum_{k=1}^{\beta-1} \tilde{c}(x_k, x_{k+1})} \cdot \delta_{x_1, s}
$$

$$
\underset{(x_{1:\beta} \models \phi)}{=} Z_{\max}^{-(\beta-1)} \cdot e^{-\sum_{k=1}^{\beta-1} c(x_k, x_{k+1})}
$$

Hence, the MAP objective is exactly equivalent to minimizing the total cost:

$$
\underset{x_{1:\beta} \in S^\beta, \, x_\beta = t}{\arg\min} -\log P(x_{1:\beta}) = \underset{x_{1:\beta} \in V^\beta, \, x_1 = s, \, x_\beta = t, \, (x_k, x_{k+1}) \in E}{\arg\min} \sum_{k=1}^{\beta-1} c(x_k, x_{k+1}),
$$

recovering precisely the original CSP objective.

**Step 5: Complexity and Conclusion.** The construction described above clearly runs in polynomial time (adding one dummy state per vertex and computing simple exponentials). Thus, we have demonstrated a polynomial-time reduction from the CSP problem (known NP-hard) to the MAP inference problem under unary constraints in a Markov chain. $\qquad\square$

## B    HARDWARE FOR EXPERIMENTS

All the experiments were run on a machine equipped with the following infrastructure: Intel® Xeon® Silver 4416+ CPU, 20 Cores, 40 Threads, 2.00/3.90 GHz; NVidia L40S GPU, 48 GB GDDR6, 18176 CUDA Cores, 142 RT Cores, 568 Tensor Cores.

## C    ORDERED FASHION MNIST

For this task, we trained a CNN model on the Fashion MNIST training set for 7 epochs using a batch size of 64 and a learning rate of 0.002. The model consists of two convolutional blocks followed by three fully connected layers, incorporating Dropout and Layer Normalization. The convolutional layers use 2×2 and 5×5 filters, each followed by MaxPooling operations. The final classification layer applies a Softmax activation for 10-class classification. Afterward, we also trained the CNN-LSTM model, where the CNN component was kept frozen and only the LSTM was trained on the Ordered Fashion MNIST training set. We selected the LSTM checkpoint that achieved the best performance on the validation set to capture the temporal dependencies across sequences. The LSTM module was implemented with a hidden dimension of 128, one recurrent layer, and batch-first input formatting. Its outputs were then passed through a fully connected linear layer mapping to the 10 output classes, enabling sequence-level classification. This setup allowed the model to leverage pre-trained spatial feature representations from the CNN while adapting the temporal modeling capacity of the LSTM to the sequential structure of the Ordered Fashion MNIST dataset. We used CrossEntropyLoss as the loss function, and the Adam optimizer for both trainings.

### C.1    NATURAL LANGUAGE CONSTRAINTS

We report here all the constraints enforced on the Ordered Fashion-Mnist dataset, expressed in natural language. The LTL$_f$ formulation of each constraint is reported below.

- If you wear a T-shirt/top, you will not be able to wear a Shirt, Dress, or another T-shirt/top.
- If you wear Trousers, you will not be able to wear a Dress or another Trouser.
- If you wear a Pullover, you will not be able to wear a Dress, T-shirt/top, Shirt, or another Pullover.
- If you wear a Dress, you will not be able to wear a T-shirt/top, Shirt, Trouser, Pullover, or another Dress.
- If you wear a Coat, you will not be able to wear a T-shirt/top, Shirt, Pullover, Dress, or another Coat.
- If you wear Sandals, you will not be able to wear Sneakers, Trousers, Ankle boots, or another pair of Sandals.
- If you wear a Shirt, you will not be able to wear a T-shirt/top, Dress, or another Shirt.
- If you wear Sneakers, you will not be able to wear Sandals, Trousers, Ankle boots, or another pair of Sneakers.
- If you wear a Bag, you will not be able to wear a T-shirt/top, Shirt, Dress, Pullover, Coat, or another Bag.
- If you wear Ankle boots, you will not be able to wear Sandals, Trousers, Sneakers, or another pair of Ankle boots.
- You must wear at least one of the following: T-shirt/top, Pullover, Shirt, or Dress.
- You must wear at least one of the following: Trouser or Dress.
- You must wear at least one of the following: Sandal, Sneaker, or Ankle boot.

## C.2 LTL$_f$ CONSTRAINTS

The listed formulas must be interpreted as being conjoined via the logical AND operator ($\wedge$). That is, the complete specification is given by:

$$\varphi = \varphi_1 \wedge \varphi_2 \wedge \cdots \wedge \varphi_{13}$$

where each $\varphi_i$ corresponds to the respective in the list:

- $G(t\_shirt\_top \rightarrow \neg F\,Shirt) \wedge G(t\_shirt\_top \rightarrow \neg F\,Dress) \wedge G(t\_shirt\_top \rightarrow WX\,G\,\neg t\_shirt\_top)$

- $G(Trouser \rightarrow \neg F\,Dress) \wedge G(Trouser \rightarrow WX\,G\,\neg Trouser)$

- $G(Pullover \rightarrow \neg F\,Dress) \wedge G(Pullover \rightarrow \neg F\,t\_shirt\_top) \wedge G(Pullover \rightarrow \neg F\,Shirt) \wedge G(Pullover \rightarrow WX\,G\,\neg Pullover)$

- $G(Dress \rightarrow \neg F\,t\_shirt\_top) \wedge G(Dress \rightarrow \neg F\,Shirt) \wedge G(Dress \rightarrow \neg F\,Trouser) \wedge G(Dress \rightarrow \neg F\,Pullover) \wedge G(Dress \rightarrow WX\,G\,\neg Dress)$

- $G(Coat \rightarrow \neg F\,t\_shirt\_top) \wedge G(Coat \rightarrow \neg F\,Shirt) \wedge G(Coat \rightarrow \neg F\,Pullover) \wedge G(Coat \rightarrow \neg F\,Dress) \wedge G(Coat \rightarrow WX\,G\,\neg Coat)$

- $G(Sandal \rightarrow \neg F\,Sneaker) \wedge G(Sandal \rightarrow \neg F\,Trouser) \wedge G(Sandal \rightarrow \neg F\,Ankle\,boot) \wedge G(Sandal \rightarrow WX\,G\,\neg Sandal)$

- $G(Shirt \rightarrow \neg F\,t\_shirt\_top) \wedge G(Shirt \rightarrow \neg F\,Dress) \wedge G(Shirt \rightarrow WX\,G\,\neg Shirt)$

- $G(Sneaker \rightarrow \neg F\,Sandal) \wedge G(Sneaker \rightarrow \neg F\,Trouser) \wedge G(Sneaker \rightarrow \neg F\,Ankle\,boot) \wedge G(Sneaker \rightarrow WX\,G\,\neg Sneaker)$

- $G(Bag \rightarrow \neg F\,t\_shirt\_top) \wedge G(Bag \rightarrow \neg F\,Shirt) \wedge G(Bag \rightarrow \neg F\,Dress) \wedge G(Bag \rightarrow \neg F\,Pullover) \wedge G(Bag \rightarrow \neg F\,Coat) \wedge G(Bag \rightarrow WX\,G\,\neg Bag)$

- $G(Ankle\,boot \rightarrow \neg F\,Sandal) \wedge G(Ankle\,boot \rightarrow \neg F\,Trouser) \wedge G(Ankle\,boot \rightarrow \neg F\,Sneaker) \wedge G(Ankle\,boot \rightarrow WX\,G\,\neg Ankle\,boot)$

- $F(t\_shirt\_top \vee Pullover \vee Shirt \vee Dress)$

- $F(Trouser \vee Dress)$

- $F(Sandal \vee Sneaker \vee Ankle\,boot)$

# D ORDERED COMMONGEN

Each experiment in Ordered CommonGen has a prompt and a set of ordered concepts, given as a constraint on the output. A sample prompt for experiments is reported below.

## D.1 PROMPT

# Instruction

Given several concepts (i.e., nouns or verbs), write a short and simple sentence that contains *all* the required words in the given order.
The sentence should describe a common scene in daily life, and the concepts should be used in a natural way.

# Examples

## Example 1
- Concepts: "dog, frisbee, catch, throw"
- Sentence: The dog eagerly chased the frisbee trying to catch it after its owner threw it.

## Example 2
- Concepts: "apple, place, tree, pick"
- Sentence: I found an apple in a place near a tree and I picked it up.

# Your Task

- Concepts: **Concepts**
- Sentence:

### D.2 LTL$_f$ CONSTRAINTS EXAMPLE

The listed formulas should be interpreted as being conjoined via the logical AND operator ($\wedge$). Therefore, the complete specification is given by:

$$\varphi = \varphi_1 \wedge \varphi_2 \wedge \cdots \wedge \varphi_5$$

where each $\varphi_i$ corresponds to the respective formula in the list below. This represents an example with 3 concepts.

- $((\neg(secondword \vee dot) \; U \; firstword) \wedge F(secondword))$
- $((\neg(thirdword \vee dot) \; U \; secondword) \wedge F(thirdword))$
- $((\neg(eos \vee dot) \; U \; thirdword) \wedge F(eos))$
- $G(dot \to X \; eos)$
- $G(firstword \vee secondword \vee thirdword \vee dot \vee eos \vee nomatch)$

### D.3 QUALITATIVE EXAMPLES

GPT-2 large on CommonGen (Concepts: "shave", "look", "mirror", "face")

No `ABS`: "The boy looks at himself in the mirror." (Fails constraint: "shave" and "face" are missing)
`ABS` ($\alpha = 0.25$): "Shave your face and look at the mirror." (Satisfies constraint, natural structure)
`ABS` ($\alpha = 0.5$): "The boy looks at the mirror to shave his face." (Satisfies constraint, natural structure)
`ABS` ($\alpha = 0.75$): "Look mirror face shave." (Satisfies constraint, but unnatural structure)

### D.4 ABLATION STUDIES

In Table 5 and 6 , we present the ablation studies to empirically demonstrate how the Ramping Push Up (RPU) is necessary to ensure constraint satisfaction. Furthermore, it can be observed that text quality metrics slightly improve when RPU is applied, while efficiency increases significantly thanks to solutions being found in fewer steps compared to standard generation when setting $\alpha > 0$.

### D.5 USE OF LARGE LANGUAGE MODELS

Large Language Models (LLMs) were employed in a limited manner during the preparation of this manuscript. They were used to assist in improving the phrasing of certain passages and to accelerate technical aspects of the writing process, such as generating LaTeX code for tables and formatting. Importantly, the use of LLMs was restricted to supporting and refining the writing process; they did not contribute to the research design, analysis, or interpretation of results. All text and suggestions generated by the models were thoroughly reviewed and verified by the authors to ensure accuracy and appropriateness before inclusion in the final version of the paper.

Table 5: Ablation Study on CommonGen (Supervised Model) with/without Ramping Push-up (RPU).

| RPU | ROUGE-L | BLEU-4 | CIDEr | SPICE | Constraint (%) | Avg Time (s) |
|-----|---------|--------|-------|-------|----------------|--------------|
| False | 49.25 | 48.85 | 15.76 | 29.61 | 99.98 | 10.01 |
| True | **49.70** | **49.50** | **16.20** | **29.70** | **100.00** | **4.42** |

Table 6: Ablation Study on CommonGen (Unsupervised Model) with/without Ramping Push-up (RPU).

| RPU | ROUGE-L | BLEU-4 | CIDEr | SPICE | Constraint (%) | Avg Time (s) |
|-----|---------|--------|-------|-------|----------------|--------------|
| False | 48.61 | 47.80 | 15.56 | 29.12 | 99.98 | 9.93 |
| True | **48.70** | **47.90** | 15.70 | **29.30** | **100.00** | **4.92** |

