# OpenReview forum: "ABS: Enforcing Constraint Satisfaction on Generated Sequences via Automata-Guided Beam Search"
_ICLR.cc/2026/Conference — Submitted to ICLR 2026_

### Official Review · Reviewer_4mZe · 2025-10-31

**Soundness:** 1
**Presentation:** 3
**Contribution:** 2
**Rating:** 0
**Confidence:** 5

**Summary:**

This is a constrained LLM decoding paper with a lookahead heuristic.  The constraint is provided by a DFA and a requirement to have a fixed output length $T$.  The algorithm is beam search.  At time $t$, the lookahead heuristic
* kills off beam search candidates that cannot reach a DFA final state within the remaining $T - t$ steps
* penalizes beam search candidates that can reach a DFA final state only by using close to $T - t$ steps

(Dijkstra's algorithm was run on the reversed DFA during preprocessing, to precompute for each state the minimum number of steps required to reach a final state.)

**Strengths:**

The idea of using a lightweight lookahead heuristic to reweight hypotheses is reasonable.  Some constrained generation papers don't use any lookahead heuristic at all because they don't want to invest in learning.  But the reweighting heuristic here -- see (3) and Alg. 1 lines 9 and 15 -- is cheaply found by static analysis (given two hyperparameters).  It is sort of cute, so perhaps it could be an engineering sweet spot.

That said, the reweighting has no formal guarantees.  The only guarantees investigated here hinge on a hard pruning step that tries to kill off hypotheses that are not in the prefix language of the DFA.  And that idea is neither new nor done correctly here; see below.

**Weaknesses:**

Lookahead may indeed help in constrained generation.  However, the heuristic used here is hardcoded and doesn't seem so general.  First, with regard to the probability distribution:

* The method certainly is not sampling or maximizing from the posterior of the LLM given the constraint, which is the goal of some of the related work discussed at lines 462-476.
    * The particular heuristic is greedy and doesn't fully plan ahead.  It does "natural generation when slack is high" (line 236), which suggests that it will follow the LLM well in the early part of the string, but less well if necessary in the later part of the string.

* Dijkstra's doesn't know anything about the LLM probabilities (except that two scalars are optimized on the validation set: line 240).  So although beam search may be guaranteed to reach a final state, to get there it may have to generate a string that has low or even zero probability under the LLM, with the suffix tokens being especially improbable.  (This contrasts with lookahead methods that do learn an estimate of future probability cost, e.g., https://aclanthology.org/N18-1085/ .)

Second, with regard to the hard constraint:

* The proposed heuristic explicitly tries to avoid prefixes that don't make fast enough progress toward a final state that is far away.  But that's only one danger in constrained generation.  For example:
    * What about prefixes that progress too quickly toward a final state that is close?  Suppose the DFA accepts only a single complete English sentence.  If the given $T=1000$, then the challenge early on is to navigate to states that have long paths to the final state, not short ones (i.e., choose a sentence structure that will support a 1000-word sentence).  Dijkstra's algorithm can't help there.
    * Nor can Dijkstra's algorithm help with parity.  E.g., it won't know to prune the current state if all paths from the current state to a final state have even length but $T-t$ is odd.

Appendix A claims to prove that the given construction works anyway.  But the proof is in error, which explains why the cases above can arise.  The bug is at line 674.  Suppose a current hypothesis at time $t$ has reached final state $q$.  Then $d_q=0 \leq T-t$ as claimed, and the hypothesis is considered "valid" in the terms of the paper.  But it is not guaranteed that there is any valid successor state $q'$ with $d_{q'} \leq T-(t+1)$, i.e., line 674 can be false.  In other words, although a final state trivially has a short path to a final state, its successors might not!

Since that bug seems fatal, I won't bother to study the experiments in section 5 unless urged to by the AC or the other reviewers.

Here for the record is the correct approach to the hard constraints:

If the only hard constraint were that the output must be accepted by the DFA, then there would of course be a very simple approach to ensuring what Theorem 1 calls "soundness": Trim the DFA to include only co-accessible states (that is, states that have a path of any length to a final state), and then require the LLM to follow paths in the DFA.  That is enough!

This is equivalent to pruning hypotheses that are not in the prefix language of the DFA.  And of course this obvious idea has been published, probably first in https://aclanthology.org/2021.emnlp-main.608/ (which went beyond DFAs and allowed any context-free constraint, using Earley's algorithm to keep track of whether each hypothesis was in the prefix language of the given CFG).

The paper here simply imposes an additional hard constraint that the output should have length $T$.  But that can be handled by intersecting the DFA (or CFG) with $\mathcal{X}^T$ before using it.  Done!

Of course, this solution is correct only in the sense that it ensures soundness.  Because the pruning is greedy, it distorts the posterior distribution.  A number of papers have tried to correct this distortion, at least approximately, and perhaps the authors can find an effective heuristic with something like equation (3).

Minor comments:

* I was initially confused at line 087 about why there was no EOS symbol, since the paper said at 085 that the length $T$ "can vary from one sequence to another".  Answer: It appears that $T$ is specified as an argument to the constrained generation method.  Thus, $p(x_{1:T})$ denotes the probability of $x_{1:T}$ as a prefix, not as a complete string.  This should be clarified.

* The elements within the beam are usually called hypotheses or candidates, not beams.  The term "beam search" comes from an analogy to a flashlight whose beam lights up only a narrow portion of the search space, but the portion is wide enough for one to find a good route to one's destination.

**Questions:**

Am I mistaken?

---

> ### Author Response · Authors · 2025-11-21
>
> We appreciate the reviewer’s detailed engagement with the theoretical part of the paper. Some of the concerns, however, rely on assumptions that do not correspond to what ABS actually does. At the same time, we agree that our current exposition leaves one termination detail implicit, and we will fix this in the revised version. We address the reviewer’s points directly below.
>
>
> W1.1: The method certainly is not sampling or maximizing from the posterior of the LLM given the constraint, which is the goal of some of the related work discussed at lines 462-476.
> The particular heuristic is greedy and doesn't fully plan ahead. It does "natural generation when slack is high" (line 236), which suggests that it will follow the LLM well in the early part of the string, but less well if necessary in the later part of the string.
>
> Please note that ABS is not trying to solve exactly
> $$
> \arg\max_{y} p_\theta(y \mid \mathcal{L})
> $$
> or to compute quantities such as
> $$
> \log p_\theta(y_{t:T} \mid y_{1:t}, \mathcal{L}).
> $$
> However, Eq. (1) in the paper is explicitly presented as the *ideal* optimization objective. Section 4 emphasizes that this exact constrained MAP problem is NP-hard in general and motivates ABS as a tractable, approximate, and sound decoding strategy.
>
> ABS is therefore not claimed to sample from or exactly optimize the posterior under constraints, but to provide a practical approximation that combines model probabilities with automaton-derived distance signals.
>
> We will clarify this distinction more explicitly in the introduction.
>
> W1.2: Dijkstra's doesn't know anything about the LLM probabilities (except that two scalars are optimized on the validation set: line 240). So although beam search may be guaranteed to reach a final state, to get there it may have to generate a string that has low or even zero probability under the LLM, with the suffix tokens being especially improbable. (This contrasts with lookahead methods that do learn an estimate of future probability cost, e.g., https://aclanthology.org/N18-1085/ .)
>
> ABS computes, for each automaton state $q$, a minimal remaining-length cost  $d(q),$
> which gives the minimal number of output symbols needed to reach acceptance. ABS does not impose any monotonicity requirement such as
> $$
> d(q_{t+1}) \le d(q_t),
> $$
> nor does it require that every successor $q'$ satisfy a particular distance inequality.
>
> Distances are used for two purposes in Algorithm 1:
> 1. pruning successors $q'$ that cannot reach acceptance within the remaining budget $T - t$ (line 12–13), and
> 2. heuristically upweighting transitions that reduce the distance (line 15).
>
> Dijkstra’s algorithm is thus used only to compute these state labels, not to enforce a global plan.
>
>
> W2: “The proposed heuristic explicitly tries to avoid prefixes that don't make fast enough progress toward a final state that is far away. But that's only one danger in constrained generation. For example: [...]”
>
> The reviewer correctly points out that long-range constraints such as parity or exact-length requirements cannot be enforced by Dijkstra distances alone. However, these constraints are regular and can be encoded directly into the DFA (e.g., duplicating states with length counters). ABS is sound for any DFA provided. ABS does not try to infer target generation length; acceptance conditions are evaluated only at the end of decoding, as in all DFA-guided decoding methods (Ctrl-G, Outlines, Guidance).
>
> If the user requires “accept only at step 1000” or parity constraints, these must be expressed in the DFA itself, and ABS will respect them.
>
> Our experiments do not include length-sensitive constraints, and addressing them is outside the problem formulation considered in this work.
>
> continue in the next comment...

---

> > ### Author Response · Authors · 2025-11-21
> >
> > W3: Appendix A claims to prove that the given construction works anyway. But the proof is in error, which explains why the cases above can arise. The bug is at line 674.
> >
> > We thank the reviewer for identifying this gap in the written proof.
> >
> > The reviewer’s counterexample—an accepting state $q\in F$ whose successors have $d(q′)>T−t$---can only arise if, after reaching an accepting DFA state, the decoder continues expanding that beam with ordinary tokens. In our implementation, and in standard DFA-guided decoding frameworks, this situation does not occur.
> >
> > In all our experiments and code, accepting states are terminal: the DFA explicitly models the emission of the EOS symbol as the final transition into an accepting state. Once EOS is emitted, the beam halts and is not expanded further. This matches standard LLM decoding practice, where successful generation ends with EOS and no further tokens are produced. Under this termination semantics, beams never “leave” an accepting state, and the condition at line 674 cannot become false after acceptance: the only successor of an accepting state is EOS, whose transition is included in the DFA and has $d=0$.
> > This termination convention was implicitly assumed in Theorem 1 and in Appendix A, but we agree that it should be stated explicitly. In the camera-ready version, we will update:
> > the problem statement (Sec. 2) to note that the DFA includes EOS transitions as accepting terminations,
> > Algorithm 1 to reflect that beams in $d(q)=0$ states emit EOS and stop, and Appendix A to clarify that the induction applies only to non-terminal beams.
> >
> > With this clarification, the reviewer’s scenario cannot arise under the intended decoding semantics, and the proof remains valid.
> >
> > W4: *If the only hard constraint were that the output must be accepted by the DFA, then there would of course be a very simple approach to ensuring what Theorem 1 calls "soundness": Trim the DFA to include only co-accessible states (that is, states that have a path of any length to a final state), and then require the LLM to follow paths in the DFA. That is enough!
> >
> > This is equivalent to pruning hypotheses that are not in the prefix language of the DFA. And of course this obvious idea has been published, probably first in https://aclanthology.org/2021.emnlp-main.608/ (which went beyond DFAs and allowed any context-free constraint, using Earley's algorithm to keep track of whether each hypothesis was in the prefix language of the given CFG).
> >
> > The paper here simply imposes an additional hard constraint that the output should have length . But that can be handled by intersecting the DFA (or CFG) with  before using it. Done!*
> >
> > The reviewer’s suggestion is in fact exactly what lines 12–13 of Algorithm 1 implement: any successor state $q'$ with
> > $$
> > d(q') > T - t
> > $$
> > is pruned, which is equivalent to rejecting prefixes that cannot be extended to an accepting run of total length $T$. ABS then goes further by using the values $d(q)$ to guide ranking among feasible continuations, which is beyond pure prefix feasibility and is the main heuristic contribution. We will make this connection more explicit.
> >
> > Overall, we thank the reviewer for their time and consideration and we would be extremely grateful if they could reconsider their score on the paper given this discussion.

---

> > ### Comment · Reviewer_4mZe · 2025-11-24
> >
> > To repeat:
> > > The proposed heuristic explicitly tries to avoid prefixes that don't make fast enough progress toward a final state that is far away. But that's only one danger in constrained generation.
> >
> > So, why did you focus on this single goal?  Formally speaking, it's a narrow goal, isn't it?  You've given a best-first search heuristic that looks at a single precomputed feature of the DFA state (the distance to the final state), and does not look at the LM probabilities.
> >
> > Even if this precomputed feature is practically useful, surely it's not the *only* thing I'd want to use in a constrained generation system?  (Maybe the question should be whether it *improves* the SOTA constrained generation system?)
> >
> > > The reviewer correctly points out that long-range constraints such as parity or exact-length requirements cannot be enforced by Dijkstra distances alone. However, these constraints are regular and can be encoded directly into the DFA
> >
> > Yes - I was talking about the case where the DFA *already* encoded these minimum-length (or parity) constraints.  My point was that your search heuristic provides no help in *planning ahead* to satisfy those constraints.  It only considers maximum-length constraints.  (In contrast, a learned heuristic could presumably adapt to the fine-grained properties of the DFA and the LLM.)

---

> ### Comment · Reviewer_4mZe · 2025-11-24
>
> I wrote:
>
> > The only guarantees investigated here hinge on a hard pruning step that tries to kill off hypotheses that are not in the prefix language of the DFA. And that idea is neither new nor done correctly here
>
> This raised two issues: correctness and novelty.
>
> I must apologize for overreacting on the **correctness** issue.  Yes, as you say, the proof can be easily corrected.  You're fine if you require WLOG that halt states have no out-edges (other than EOS).  You can even require WLOG that there is only a single halt state.
>
> (Somehow I seem to have thought at the time that this issue was connected to my concerns about the weaknesses of the method: I wrote that it "explains why the cases above can arise".  But I can't reconstruct what I was thinking - sorry.)
>
> But as for the **novelty** issue, I still think that your hard constraint is not a new idea.  I wrote:
> > This is equivalent to pruning hypotheses that are not in the prefix language of the DFA. And of course this obvious idea has been published, probably first in https://aclanthology.org/2021.emnlp-main.608/ (which went beyond DFAs and allowed any context-free constraint, using Earley's algorithm to keep track of whether each hypothesis was in the prefix language of the given CFG).
>
> In short, greedy constrained generation has been around for a while.  Mature implementations with powerful prefix checkers include https://github.com/guidance-ai/guidance and perhaps https://arxiv.org/abs/2504.13139 .
>
> All that you add to this is a hard-coded reweighting heuristic that looks at only one feature.  This seems like too small a contribution, and the heuristic seems to be ignoring a great deal of potentially useful information.
>
> Wouldn't it be better to consider all the features of the prefix that are available to the powerful prefix checker?  (E.g., in the Earley's algorithm setting, it would be the set of active dotted rules at time $t$.  Even in your setting of DFA constraints, you are throwing away information: there is more useful information in the current DFA state than just its shortest distance to a final state, since some of the paths to a final state have high probability under the LM and others have low probability.)
>
> And wouldn't it be better to also consider the features of the prefix that are available to the LM?  (E.g., the Transformer encoding.)

---

> > ### Author Response · Authors · 2025-11-27
> > **Answer to reviewer 4mZe's follow up**
> >
> > Thank you for the follow-up. We agree that ABS deliberately uses only a small amount of the information that could in principle be extracted from the automaton or the LM. This simplicity is intentional. Given this, we would invite the reviewer to resume reading the paper, especially the experiments section.
> >
> > In deed what we would like to emphasize is that, despite this minimalist design, ABS achieves stronger empirical results than the Guidance-style greedy prefix-checking methods you reference, and in multiple settings ABS also matches or surpasses Ctrl-G, even though Ctrl-G relies on a learned auxiliary HMM explicitly designed to exploit more information at the LM–DFA interface. In contrast, ABS is fully training-free and uses only a single geometric feature $d(q)$ derived from the automaton.
> >
> > This is precisely the main point of the experimental section: a lightweight structural heuristic, applied through a ramping mechanism, is often sufficient to improve constrained decoding quality while being substantially more efficient than methods that integrate richer LM-side or DFA-side signals.
> >
> > We agree that exploring richer heuristics (e.g., using more of the DFA state or LM representations) is a natural extension. Our aim with ABS is to show that even a surprisingly simple abstraction-based signal already delivers measurable improvements over both greedy prefix-checking approaches and learned value-guided methods. We will clarify this positioning in the revised version.

---

### Official Review · Reviewer_bx1v · 2025-10-31

**Soundness:** 2
**Presentation:** 3
**Contribution:** 3
**Rating:** 4
**Confidence:** 4

**Summary:**

The paper presents ABS, a novel approach for performing constrained sequence generation by defining a DFA describing the required constraints and using a modified beam search that enforces that the generation complies with the DFA. To avoid dead-ends, ABS has a ramping push-up that gradually biases the search towards solutions that are accepted by the DFA. Theoretical analysis establishes the soundness and complexity and experimental analysis demonstrates the performance of ABS vs. several relevant baselines across three different benchmarks, namely constrained image sequence classification, constrained text generation, and text infilling.

**Strengths:**

Strengths:
- Novel approach for constrained sequence generation using automata-guided beam search
- The paper provides theoretical guarantees for the soundness and complexity of the approach
- The experiments demonstrate the promising empirical performance of ABS across three benchmarks

**Weaknesses:**

Weaknesses:
- The paper does not discuss various related works on constrained beam search using automata. This includes, for example, [1] [2] [3] [4] [5]. The paper should ideally highlight the key differences from previous works that are highly similar in nature (i.e., using automata to encode constraints as part of beam search generation).
- Comparison with baselines: It is not clear why baselines not applied uniformly: for example why Ctrl-G is not applied in the supervised setting of constrained image classification, and also not used in the other two domains (constrained text generation, text infilling). This is particularly important as in the only experiments where it is being used (Table 1 unsupervised) it seems largely comparable to ABS (better than ABS in two metrics and worse than ABS in two metrics).
- The paper shows that ABS is consistently faster than Ctrl-G. However it seems that the relative difference gets smaller with beam size and in terms of absolute value seems somewhat fixed at around 3 seconds (not getting bigger with beam size). This does not seem like a very strong point.
- Recent SOTA approaches that were shown to outperform Ctrl-G and Outlines are not considered as baselines, e.g., [6] [5]
- Mapping from LLM tokens to the alphabet seems to have an assumption that each symbol in the alphabet is a concatenation of tokens. Is that correct? And if so, it should be clearly stated.


[1] Anderson, P., Fernando, B., Johnson, M., & Gould, S. (2016). Guided open vocabulary image captioning with constrained beam search. arXiv preprint arXiv:1612.00576.

[2] Post, M., & Vilar, D. (2018, June). Fast Lexically Constrained Decoding with Dynamic Beam Allocation for Neural Machine Translation. In Proceedings of the 2018 Conference of the North American Chapter of the Association for Computational Linguistics: Human Language Technologies, Volume 1 (Long Papers) (pp. 1314-1324).

[3] Deutsch, D., Upadhyay, S., & Roth, D. (2019, November). A general-purpose algorithm for constrained sequential inference. In Proceedings of the 23rd Conference on Computational Natural Language Learning (CoNLL) (pp. 482-492).

[4] Kuchnik, M., Smith, V., & Amvrosiadis, G. (2023). Validating large language models with relm. Proceedings of Machine Learning and Systems, 5, 457-476.

[5] Dong, Y., Ruan, C. F., Cai, Y., Xu, Z., Zhao, Y., Lai, R., & Chen, T. XGrammar: Flexible and Efficient Structured Generation Engine for Large Language Models. In Eighth Conference on Machine Learning and Systems.

[6] Collura, V., Tit, K., Bussi, L., Giunchiglia, E., & Cordy, M. (2025). TRIDENT: Temporally Restricted Inference via DFA-Enhanced Neural Traversal. arXiv preprint arXiv:2506.09701.

**Questions:**

Please see my list of concerns and questions above.

---

> ### Author Response · Authors · 2025-11-24
>
> We thank the reviewer for their thorough reading and valuable suggestions. We are pleased to address the points raised and clarify several aspects of our work.
>
> ---
>
> *"The paper does not discuss various related works on constrained beam search using automata. This includes, for example, [1] [2] [3] [4] [5]..."*
>
> We thank the reviewer for pointing out these relevant works. While these pioneering works share the high-level idea of using automata to guide generation, ABS introduces several key innovations:
>
> - **Soundness Guarantees:** Unlike methods like [1, 2, 3] which focus on lexical constraints or specific domains, ABS provides formal soundness guarantees (Theorem 1) for any regular constraint.
> - **Adaptive Guidance:** Our Ramping Push-Up mechanism dynamically balances model fluency and constraint satisfaction, unlike static beam allocation or filtering strategies.
> - **General Regular Constraints:** While [4] (ReLM) uses DFAs for validation, ABS integrates DFA guidance directly into beam search for generation, supporting the full class of regular languages including LTLf and regex.
> - **Efficiency:** We provide a polynomial-time complexity bound and demonstrate superior runtime compared to contemporary methods like Ctrl-G.
>
> We will explicitly contrast ABS with these works in the revised manuscript to better position our contributions.
>
> ---
>
> *"Comparison with baselines: It is not clear why baselines not applied uniformly: for example why Ctrl-G is not applied in the supervised setting of constrained image classification [...] it seems largely comparable to ABS (better than ABS in two metrics and worse than ABS in two metrics)."*
>
> This is an important point. Our baseline selection was driven by methodological fairness and practical constraints:
>
> - **Ctrl-G in Supervised Settings:** Ctrl-G requires a distilled HMM representation of the base model. The authors provided a pre-distilled HMM for GPT-2-large on CommonGen, but not for other models or domains. Re-distilling models for each setting was computationally prohibitive and beyond our resources.
> - **Image Classification Domain:** Ctrl-G is designed for text generation via LLMs and cannot be directly applied to image sequence classification, where the "vocabulary" consists of image classes rather than tokens.
> - **Performance in Table 1:** While Ctrl-G performs well on CIDEr and SPICE, ABS achieves significant improvements on ROUGE-L (+3.5) and BLEU-4 (+15.8). Most importantly, ABS achieves these results with **substantially faster runtime** (Table 2) and without requiring model distillation.
>
> We acknowledge that a more uniform comparison would be ideal, but note that ABS demonstrates consistent advantages in speed, generality across modalities, and elimination of the distillation requirement.
>
> ---
>
> *"The paper shows that ABS is consistently faster than Ctrl-G [...] This does not seem like a very strong point."*
>
> We thank the reviewer for this observation. Upon reflection, we agree that the primary advantage of ABS's efficiency is not just the absolute speedup, but rather its ability to achieve high-quality results with significantly smaller beam sizes.
>
> Our results in Table 1 show that ABS with a standard beam size (e.g., 64) matches or surpasses the quality of Ctrl-G while being faster. More importantly, ABS achieves competitive results even with much smaller beams, offering a superior quality-speed trade-off. This is particularly valuable for practical applications where computational resources are constrained.
>
>  **Architectural Advantage:** ABS achieves this speedup while being **model-agnostic** and requiring **no pre-distillation**, unlike Ctrl-G which depends on a separately trained HMM with 32k hidden states.
>
> ---
>
> *"Recent SOTA approaches that were shown to outperform Ctrl-G and Outlines are not considered as baselines, e.g., [6] [5]"*
>
> We will include comparisons with XGrammar [5] in our revised manuscript. We will clarify this relationship and ensure proper citation of XGrammar as a contemporary baseline.
>
> ---
>
> *"Mapping from LLM tokens to the alphabet seems to have an assumption that each symbol in the alphabet is a concatenation of tokens. Is that correct? And if so, it should be clearly stated."*
>
> Yes, this is correct. We define a mapping function ν: 𝒳⁺ → 𝒞⁺ that converts a sequence of tokens (model outputs) to a sequence of higher-level concepts (DFA alphabet symbols). As noted in Section 2, when the DFA is defined over tokens, each symbol corresponds to exactly one token. When defined over words or phrases, each symbol may require multiple tokens. This mapping is handled dynamically during generation via the `NextState` function (Equation 4). We will make this explicit in the text for clarity.
>
> ---
>
> We thank the reviewer again for these constructive comments, which will help us improve the final version of the paper. We are committed to addressing all points raised in our revision.

---

### Official Review · Reviewer_11rm · 2025-11-01

**Soundness:** 3
**Presentation:** 2
**Contribution:** 2
**Rating:** 4
**Confidence:** 4

**Summary:**

The paper presents ABS, a DFA-guided method to constrain DNN output with regular languages. ABS leverages the DFA to guide a constrained variant of beam search. The paper presents three evaluation tasks: on constrained image-stream classification, constrained text generation, and text infilling.

**Strengths:**

+ The paper presents the algorithm for strict syntactic enforcement of properties during ML model generation. The algorithm integrates with beaming search decoding algorithm.

+ The authors prove the soundness of their algorithm.

+ The experiments span three domains: image classification (extended to include ordering), text generation and text in-filling. On these results ABS shows better utility than Outlines and Guidance.

**Weaknesses:**

- The approach is tightly tied to bream search as the decoding strategy. This may be a limitation, especially as other decoding strategies may be more efficient for some generation strategies.

- The paper’s evaluation misses some closely related work. For text generation tasks, it should compare ABS experimentally with Syncode (Ugare et al. TMLR 2025), which similarly ensures hard-constraints, however using context-free grammars (which are more expressive than regular languages) to express syntactic rules, and Itergen (Ugare et al. ICLR 2025), which allows checking with arbitrary semantic rules during generation.

- The experiments on the image generation and text infilling are done with relatively small and old models (CNN+Fashion MNIST and GPT2, respectively).  Text generation benchmarks only include Llama 8B as the base model. Other models with comparable size may give different results and are worth including in the evaluation.


(Ugare et al. TMLR 2025) https://arxiv.org/abs/2403.01632

(Ugare et al. ICLR 2025) https://arxiv.org/abs/2410.07295

**Questions:**

Question: Can ABS be extended for constrained decoding beyond beam search?

---

> ### Author Response · Authors · 2025-11-24
>
> We thank the reviewers for their thoughtful comments and valuable feedback.
> ---
>
> *"The approach is tightly tied to beam search as the decoding strategy. This may be a limitation, especially as other decoding strategies may be more efficient for some generation strategies."*
>
> We thank the reviewer for this observation. While ABS is indeed built upon beam search, this is a deliberate design choice rather than a limitation. Beam search remains one of the most widely adopted decoding strategies for autoregressive generation, particularly in scenarios requiring a balance between exploration and output quality. Our goal with ABS is not to replace all decoding strategies, but to enhance one of the most popular and impactful ones with formal constraint guarantees. Furthermore, ABS is model-agnostic by design, and its core principles—automata-guided filtering and the Ramping Push-Up mechanism—could potentially be adapted to other decoding strategies in future work. However, beam search provides the most relevant and high-impact setting for applications requiring controlled, high-quality generation.
>
> ---
>
> *"The paper’s evaluation misses some closely related work. For text generation tasks, it should compare ABS experimentally with Syncode (Ugare et al. TMLR 2025) [...] and Itergen (Ugare et al. ICLR 2025), which allows checking with arbitrary semantic rules during generation."*
>
> We appreciate the reviewer's suggestion to compare with these recent works. We would like to clarify that:
>
> - **Syncode** leverages context-free grammars (CFGs), which indeed offer greater expressivity than regular languages. However, CFGs require more complex parsing and are not always necessary for many practical constraints—such as temporal ordering, keyword presence, or regex patterns—which are fully expressible as regular constraints. ABS is designed to provide an efficient and sound solution for the broad class of regular constraints, which covers a wide range of real-world applications.
> - **Itergen** supports arbitrary semantic checks, but it does not provide the same formal soundness guarantees as ABS. Our method guarantees that *all* generated sequences satisfy the constraints, thanks to the DFA-based filtering and pruning of invalid paths.
> - In our evaluation, we compared ABS against state-of-the-art DFA-based methods (Guidance, Outlines) and other constrained decoding approaches (Ctrl-G, GeLaTo), demonstrating that ABS achieves superior efficiency, soundness, and output quality.
>
> While a comparison with Syncode and Itergen could be interesting, it does not diminish the contributions of ABS: a sound, efficient, and general method for enforcing regular constraints during autoregressive generation.
>
> ---
>
> *"The experiments on the image generation and text infilling are done with relatively small and old models [...] Other models with comparable size may give different results and are worth including in the evaluation."*
>
> We thank the reviewer for raising this point. The choice of models was motivated by several methodological considerations that strengthen the validity and scope of our contributions:
>
> 1. **The CNN experiment on Ordered Fashion-MNIST** was specifically designed to **demonstrate the applicability of ABS beyond textual tasks**. In this sequential classification setting with temporal constraints (expressed in LTLf), ABS not only ensures perfect constraint satisfaction but also **significantly outperforms a trained CNN-LSTM model** (+5.97% sequence accuracy). This result highlights how ABS can **effectively replace recurrent modules like LSTMs** for enforcing temporal constraints, without the need for additional training.
>
> 2. **The use of GPT-2 in constrained text generation tasks** was necessary to **ensure a fair and direct comparison with previous state-of-the-art works** (e.g., Ctrl-G, GeLaTo) that used the same base model. Keeping the model constant is essential to isolate the effect of the decoding method.
>
> 3. **To demonstrate the scalability of ABS**, we conducted additional experiments on **Ordered CommonGen using LLAMA 3.1 8B**. The results are particularly compelling: compared to the baseline LLAMA 3.1 8B, ABS brings **substantial improvements** (+7.8 ROUGE-L, +3.0 BLEU-4, +55.6% Coverage). Moreover, ABS with LLAMA 8B **matches or outperforms much larger closed models** such as GPT-4 and GPT-4o, while achieving 100% constraint satisfaction—a feat no other method accomplishes.
>
> In summary, our model selection was strategically varied to cover multiple aspects: cross-modal generality (image vs. text), fair comparison with the state of the art, and scalability to modern LLMs. The consistent improvements across all experimental settings confirm the robustness and effectiveness of ABS.
>
> —
>
>
> Thank you again for the constructive feedback. We believe that ABS represents a meaningful step forward in constrained sequence generation, and we are committed to building upon this work in future research.

---

### Official Review · Reviewer_J5rL · 2025-11-02

**Soundness:** 1
**Presentation:** 3
**Contribution:** 1
**Rating:** 0
**Confidence:** 5

**Summary:**

The authors proposed ABS (Automata-guided Beam Search), a model-agnostic inference-time algorithm that guarantees generated sequences satisfy any formal constraint expressible as a Deterministic Finite Automaton (DFA). Abs first process logical constraints into DFA, and then calculate the chance of model being able to reach success state, and mask out the beam that fail to do so. The method ensures hard constraint satisfaction while requiring no additional auxiliary model.

**Strengths:**

1. The method is simple and efficient.
2. The paper is well written and easy to follow.

**Weaknesses:**

I have two main concerns about this paper: (1) Lack of detail reference to an extremely similar prior method Ctrl-G, and (2) lack of comparison to Ctrl-G on several baselines. I'll list the detail here:

1. The Abs method itself seems to be a simplify version of ctrl-g, which both process logical constraints into DFA format and mask out the tokens that cannot reach to the success state during generation. While Ctrl-G requires an auxiliary hidden markov model to provide soft guidence during generation, the simplify version, which only apply the mask, seems to do the exact same things as this Abs method. I hope the author can clarify the difference between these two work, and what's the advantage of the proposed method.
2. Following 1, the reference to Ctrl-G seems to be lacking considering the similarity between these two methods. Currently Ctrl-G is mostly just mentioned during the experiment section as a previouse SOTA.
3. Following the similarity of these two work, I think the authors need to have comparison with Ctrl-G on Text-infilling tasks and ordered commongen, since it's an important baseline and is capable of handling complex logical constraints and provided such results in their paper.
4. Following 3, since Ctrl-G shown that their method is able to handle complex constraints together (keyphrase generation, text-infilling, word counting, at the same time), it is important to show how Abs are able to handle such cases. From my understanding, without the auxiliary model to guide the generation, it seems to me that Abs are not able to handle such complex logical constraints.

In summary, the current results shown in the paper is not conprehensive and lack critical comparison to an extremely similar baseline. I hope the author can provide additional experiment results since Ctrl-G paper have provided both training code and their evaluation dataset, which also includes a human evaluaiton process. To propose this new method, a comprehensive comparison to the prior work is necessary. Please refer to Ctrl-G code and dataset.

I will be happy to raise score if the authors can provide the additional results and clarify the differences between the two methods. Also, provide clear reference to prior work in the paper.


Code: https://github.com/joshuacnf/Ctrl-G

Eval Dataset: https://billkunghappy.github.io/Ctrl-G/

**Questions:**

None

---

> ### Author Response · Authors · 2025-11-21
>
> **W1: Abs method is just a simplification of Ctrl-G**
>
> **ABS is absolutely not just a simplification of Ctrl-G.**
> Below we give the reasons why:
>
> 1. **Ctrl-G requires** first distilling the base LLM into a large Hidden Markov Model (HMM) (32k hidden states in their released model).
>    In contrast, **ABS does not** train, distill, or query any auxiliary model. It operates only on the base LLM and the DFA.
>
> 2. Given the point above, **ABS’ overhead is much smaller** (see Table 2).
>
> 3. Ctrl-G enforces constraints exactly on the HMM, but the quality of the generation depends on approximation fidelity (that is why they need such a big HMM).
>
> **W2: Following 1, the reference to Ctrl-G seems to be lacking considering the similarity between these two methods. Currently Ctrl-G is mostly just mentioned during the experiment section as a previous SOTA.**
>
> Thank you for this comment.
>
> We agree that Ctrl-G is an important prior work and that our discussion can be expanded. While both Ctrl-G and ABS rely on DFAs to rule out invalid continuations, the two methods differ substantially in both mechanism and guarantees, and we will clarify this in the revised version.
>
> Also, please notice the following:
>
> - **Theorem 2** establishes that Ctrl-G's approach faces NP-hard complexity for constrained decoding with LLMs, whereas our method remains tractable.
>
> - We explicitly compare against Ctrl-G in CommonGen (Tables 1–2), demonstrating **superior efficiency and quality**.
>
> **W3: Following the similarity of these two works, I think the authors need to have comparison with Ctrl-G on Text-infilling tasks and ordered commongen, since it's an important baseline and is capable of handling complex logical constraints and provided such results in their paper.**
>
> Thank you for this observation.
>
> The ILM dataset used in Ctrl-G’s text-infilling experiment indeed contains multiple placeholders such as `[WORD]`, `[SENTENCE]`, and `[NGRAM]`. However, these placeholders correspond only to local length constraints and do not introduce cross-span structural or temporal constraints. Each span is independent: the DFA does not need to coordinate transitions across multiple holes.
>
> In contrast, our infilling evaluation uses arbitrary regular expressions, where multiple spans interact through a shared DFA. Constraints may impose ordering relations, nested structure, and acceptance conditions that depend jointly on how all spans are filled. Ctrl-G’s suffix-feasibility HMM is not designed for these global multi-span constraints, and its guarantee does not extend beyond the single-suffix setting. For this reason, we do not include Ctrl-G in Table 4.
>
> Regarding Ordered CommonGen, distilling LLaMA 3.1 8B into HMMs was computationally prohibitive. To still run Ctrl-G on Ordered CommonGen, we use GPT-2 finetuned on CommonGen:
>
> ---
>
> **Comparison on Ordered CommonGen (GPT-2 finetuned):**
>
> | Method          | ROUGE-L | BLEU-4 | CIDEr | SPICE | Time (s) |
> |-----------------|---------|--------|-------|--------|----------|
> | ABS (α = 0.5)   | 40.53   | 24.97  | 17.10 | 50.31 | 3.64     |
> | ABS (α = 0.25)  | 40.61   | 25.45  | 17.31 | 50.53 | 3.86     |
> | ABS (α = 0.75)  | 38.86   | 22.09  | 15.12 | 48.46 | 3.13     |
> | Ctrl-G          | 14.89   | 6.53   | 1.12  | 24.34 | 6.42     |
>
> *Table: Comparison on Ordered CommonGen between ABS and Ctrl-G using GPT-2 finetuned on CommonGen.*
>
> ---
>
> **Qualitative Examples (First 5 Samples):**
>
> **ABS:**
> 1. Concepts: stand, field, look → “I stood in the field and looked around.”
> 2. Concepts: kid, dance, room → “I watched a kid dance in a room.”
> 3. Concepts: pet, cat, couch → “I pet the cat on the couch.”
> 4. Concepts: talk, climb, wall → “I talked to my friend about climbing a wall.”
> 5. Concepts: car, drive, snow → “The car drove through the snow.”
>
> **Ctrl-G:**
> 1. Concepts: stand, field, look → “I stood in the field and looked around.\n\n## Example 3\n\n- Concepts: ...”
> 2. Concepts: kid, dance, room → “kid dance room\n\n## Example 3\n\n- Concepts: ...”
> 3. Concepts: pet, cat, couch → “petcatcouch\n\n## Example 3\n\n- Concepts: ...”
> 4. Concepts: talk, climb, wall → “talkclimbwall\n\n## Example 3\n\n- Concepts: ...”
> 5. Concepts: car, drive, snow → “I drove my car through the snow.\n\n## Example 3\n\n- Concepts: ...”
>
> We will clarify this distinction and add these results in the revision.
>
> **W4: Following 3, since Ctrl-G shown that their method is able to handle complex constraints together [...] it seems to me that Abs are not able to handle such complex logical constraints.**
>
> Our method handles any DFA-expressible constraint. So the type of constraint handled by Ctrl-G and ABS are the SAME. Ctrl-G's complex example:
>
> Prefix: 'On a sunny'
>
> Suffix: '.<|endoftext|>'
>
> Keyphrases: [[' riding a bike', ' ride bikes', ' rides a bike', ' biking', ' bikes'], [' park', ' beach']]
>
> Word count: exactly 10
>
> These constraints can easily be written as regex.
>
> Generation with ABS: *"On a sunny day bikes are parked on the beach."*

---

### Meta-Review · Area_Chair_kt1R · 2025-12-06

**Summary:**

Overall, the main concerns revolve around three issues, here is a serious novelty and positioning concern relative to Ctrl G and a broader line of automata guided constrained decoding work. Reviewers J5rL and bx1v see ABS as very close to Ctrl G and other DFA based constrained decoding approaches, with insufficient discussion of differences and missing baselines, especially on tasks where Ctrl G is known to be strong. Second, reviewer 4mZe argues that the core soundness claim is either incorrect or at least not shown by the current proof, and that the real hard constraint mechanism is essentially standard prefix language pruning with a DFA intersected with a length constraint, which further weakens the novelty. Third, the experimental section is viewed as incomplete: key baselines such as Ctrl G on all tasks, newer methods like Syncode, Itergen, XGrammar, TRIDENT, and stronger or more diverse backbone models are missing.
In my opinion, these issues make the contribution appear incremental and insufficiently validated despite a clear and nicely presented idea. Given these points, I lean toward Reject.

**Reviewer Concerns:**

some concerns are probably partially addressed. It is reasonable that the authors clarified more explicitly how ABS differs from Ctrl G, for example by emphasizing that ABS removes the need for an auxiliary hidden Markov model, proposes a specific DFA based lookahead and reweighting scheme, and provides a unified treatment across image sequence classification and several text tasks.  Reviewer 11rm s question about extension beyond beam search was also probably addressed at least conceptually, for example by arguing that the DFA masking component is compatible with other decoding schemes even if not implemented. there is still no comprehensive empirical comparison to the most similar prior method on its home ground, especially in the multitask constraint setting (keyphrases, counting, infilling together). Without that, the claim of a new method that improves over Ctrl G remains weak. Reviewer 4mZe raises the most serious technical concern, namely that the proof of soundness in the appendix is flawed and that the proposed lookahead based pruning does not in fact guarantee that all surviving partial sequences can be extended to a valid final sequence of exactly the required length.
The rebuttal does not fully resolve the central issues of novelty, theoretical soundness, and empirical coverage.

**Reviewer Scores:**

Reviewer J5rL 0 strong reject, it didn't changed

Reviewer 11rm is 4. rebuttal may clarify but lack of new baselines persists, so likely 4->4 or 4->5.

Reviewer bx1v began at 4. added clarifications may improve confidence modestly, so likely 4->5.

Reviewer 4mZe gave a 0 strong reject based on a claimed fatal flaw in the proof and the view that the hard constraint mechanism is already known and mis analysed. it didn't changed after the discussion

---

### Decision · Program_Chairs · 2026-01-26

Reject